# Design and Fabrication of Pneumatically Actuated Valveless Pumps

**DOI:** 10.3390/mi13010016

**Published:** 2021-12-23

**Authors:** Jr-Lung Lin

**Affiliations:** Department of Mechanical and Automation Engineering, I-Shou University, Kaohsiung City 84001, Taiwan; ljl@isu.edu.tw

**Keywords:** valveless, pump, larger deformation, pneumatically actuated

## Abstract

In this study, a valveless pump was successfully designed and fabricated for the purpose of medium transportation. Different from traditional pumps, the newly designed pump utilizes an actuated or a deflected membrane, and it serves as the function of a check valve at the same time. For achieving the valveless property, an inlet or outlet port positioned in an upper- or lower-layer thin membrane was designed to be connected to an entrance or exit channel. Theoretical analysis and numerical simulation were conducted simultaneously to investigate the large deformation characteristics of the membranes and to determine the proper location of the inlet or outlet port on the proposed pump. Then, the valveless pump was fabricated on the basis of the proposed design. In the experiment, the maximum flow rate of the proposed pump exceeded 12.47 mL/min at a driving frequency of 5.0 Hz and driving pressure of 68.95 kPa.

## 1. Introduction

In recent decades, micropumps have been successfully designed and fabricated for a wide number of applications. Different micropumps are effective for different applications of drug delivery [1,2]. Therefore, increasingly versatile micropumps that are useful in multiple applications are being developed with high effectiveness. These devices have also been introduced to the ever-challenging field of drug delivery. For example, micropumps have been used in insulin delivery systems to maintain diabetics’ blood sugar [3,4]. Patient-controlled analgesia helps relieve patients’ serious pain after surgery. Currently, micropumps have been used as ventricular assist devices to pump blood for cases of heart failure [5,6]. Therefore, micropumps with a relatively high throughput present a challenging issue for researchers.

A simple micropump consists of a fluidic chamber, an actuator, a diaphragm, and valves. In principle, the fluidic chamber should keep the liquid fluid until the pumping action. The actuators cause the reciprocation of the diaphragm to provide the energy of the fluid flow. The valves mainly restrict the fluid flow to one direction. The reciprocating diaphragms are generally actuated by electrostatic, electromagnetic, pneumatic, thermopneumatic, or piezoelectric modes [1,2,7]. Generally, the diaphragm displacement profile is symmetric, and thus, the flow is non-directional. The oscillatory movement of the diaphragm of a micropump can cause the fluid to flow toward the inlet and outlet directions. Some arrangements for flow rectification need to be made to obtain the net flow in the desired outlet direction. The most common technique is to use valves that allow flow in one direction only.

Overall, micropumps can be divided as either non-mechanical or mechanical scheme based on whether their parts are fixed or movable [1,2,7]. According to different flow rectification methods, micropumps can have valves or be valveless. Non-mechanical pumps are also classified as valveless pumps, including electrohydrodynamic (EHD) [8], magnetohydrodynamic (MHD), electroosmotic, electrowetting, bubble type, electrochemical [9], and self-oscillating polymer gels modes [10]. The valves used in micropumps can be categorized as dynamic valves and static valves depending on the presence of moving parts. Dynamic valves are those that close and open because of external actuated energy or pressure difference, namely active [11,12,13,14] or passive valves [15,16,17,18,19,20]. The active valves required different actuated modes, including electrical [11], electrostatic [12], pneumatic [13], and piezoelectric [14], respectively. Otherwise, passive valves do not require additional external actuation for deflection or deformation. Some examples of passive valves include check valves [15,16,17], flapping valves [18,19,20], or peristaltic valves [21].

The benefits of dynamic valves in micropumps include high operating pressure, good flow direction, and minimal backflow [7,22]. However, these valves can be affected by the working fluids containing cells or particles that are prone to damage or clogging [23]. High-pressure loss at these valves may also be a curial problem. As valves are always applied under conditions with high pressure or high-pressure differences, valve materials should have adequate fatigue strength for long-term operation [24]. Many disadvantages still exist, including having a complex structure and being difficult to integrate with the microfluidic system [7]. By contrast, static valves are those with a fixed geometry [25] and are free from moving parts. The most common type of static valves is the nozzle diffuser [26,27,28,29,30]. Micropumps equipped with nozzle diffuser elements as flow rectification devices are commonly known as valveless micropumps. The advantage of nozzle diffuser elements is that no moving parts or boundaries are involved; hence, wear and mechanical and fatigue failure are effectively reduced. In addition, valveless micropumps are easy to fabricate, and they last for a long time. However, they are prone to relatively higher resistant loss and lower flow rate [28,30]. Moreover, reverse flow is a crucial problem in micropumps with nozzle diffuser elements. This complexity of reverse flow has become a challenge and thus restricts the application of valveless micropumps. Therefore, the design of the structure is important for valveless pumps. Moreover, it could be helpful to improve the performance of valveless pumps [25]. Summarily, the performance comparisons of various valveless pumps are illustrated as Table 1.

It is worth mentioning that peristaltic micropumps rectified by actuators not valves are classified as the valveless pumps. In the transporting approach, the fluid is perpendicular actuated in sequence by membranes to the flow direction. In particular, a caterpillar locomotion-inspired valveless pneumatic micropump was successfully developed to provide new opportunities for the simple integration of microfluidic systems [31]. The fluid was continuously driven by a single teardrop-shaped elastomeric membrane along the flow direction. However, the limitation of flow rate is still a crucial technique for the pumps mentioned above. Therefore, the motivation of this study is to design an easily fabricated, low-cost, less leakage, long-term working, and high volumetric flow rate valveless pump extensively for the applications of intravenous infusion.

In this study, a new valveless micropump is designed to transport media that utilizes an actuated or a deflected membrane, and it serves as the function of a check valve at the same time. Theoretical and numerical models were used to predict the deformation mechanism and to determine the proper location of the inlet/outlet ports. Moreover, the proposed valveless micropump was tested to experimentally investigate its pumping performance.

## 2. Theoretical Analysis

Over the past decade, a large number of PDMS applications have been successfully carried out in the literature. However, an approximating model was provided to evaluate the large deformation mechanism [32,33]. The deformation behavior of the PDMS membrane is still unclear. Therefore, in the current work, the deflection of the flexible membrane is theoretically analyzed to be derived from the Timoshenko model [34]. In the current model, the profile of *w**(*r**) is assumed to be the same as the previous literatures [32,33,34], but the assumption of *u**(*r**) is different.

When external energy, namely, bending and stretching strain energy, is applied to a flexible membrane, the membrane exhibits deformation. The strain energy of bending (*U_b_*) and stretching (*U_S_*) of the circular membrane can be expressed as [34]
(1)$$U_{b}=\frac{Eh^{3}}{24(1-v^{2})}\int_{0}^{2\pi }\int_{0}^{a}\left[{\left(\frac{\partial^{2}w}{\partial r^{2}}\right)}^{2}+\frac{1}{r^{2}}{\left(\frac{\partial w}{\partial r}\right)}^{2}+\frac{2v}{r}\frac{\partial w}{\partial r}\frac{\partial^{2}w}{\partial r^{2}}\right]rdrd\theta$$
(2)$$U_{S}=\frac{Eh^{3}}{24(1-v^{2})}\int_{0}^{a}\left({\left[\frac{du}{dr}+\frac{1}{2}{\left(\frac{dw}{dr}\right)}^{2}\right]}^{2}+{\left(\frac{u}{r}\right)}^{2}+\frac{2\nu u}{r}\left(\frac{du}{dr}+\frac{1}{2}{\left(\frac{dw}{dr}\right)}^{2}\right)\right)rdr.$$

Here, *u* and *w* are the radial (*r*-directional) and tangential (*z*-directional) displacements, respectively. The relationship of *u* and *w* must satisfy the following equation [34]:(3)$$\frac{d^{2}u}{dr^{2}}=-\frac{1}{r}\frac{du}{dr}+\frac{u}{r^{2}}-\frac{1-v}{2r}{\left(\frac{dw}{dr}\right)}^{2}-\frac{dw}{dr}\frac{dw^{2}}{dr^{2}}$$
(4)$$\frac{d^{3}w}{dr^{3}}=-\frac{1}{r}\frac{d^{2}w}{dr^{2}}+\frac{1}{r^{2}}\frac{dw}{dr}+\frac{12}{h^{2}}\frac{dw}{dr}\left[\frac{du}{dr}+\nu \frac{u}{r}+\frac{1}{2}{\left(\frac{dw}{dr}\right)}^{2}\right].$$

Although Equations (3) and (4) are nonlinear equations that can be calculated numerically, they are too complex to be analytically solved. Most circular diaphragms are considered to have clamped boundary conditions, that is,
(5)$$\frac{\partial w^{*}(0)}{\partial r^{*}}=0;\;\;\;w^{*}(1)=0.$$

Here, *w**(*r**) *= w*/*w_o_,* and *r** *= r*/*a. w_o_* is the characteristic displacement of *w*(*r*). *w*(*r**) is assumed to be expressed by
(6)$$w^{*}(r^{*})={\left(1-r^{*}\right)}^{n}.$$

Through the differentiation of *d**w**(*r*)/*dr** = 0, the maximum displacement of *w*_max_ can be calculated as *w_o_* at *r** = 0.

*u* should also satisfy the boundary conditions, that is,
(7)$$u^{*}(0)=u^{*}(1)=0.$$

Here, *u**(*r**) = *u*/*u*_o_, and *r** = *r*/*a*. *u_o_* is also the characteristic displacement of *u*(*r*). The comparison of Equations (3) and (4) shows that the order degree of *u*(*r*) is (2*n* − 1) higher than that of *w*(*r*) on the basis of the order of magnitude analysis. Therefore, the profile of *u**(*r**) can be assumed as
(8)$$u^{*}(r^{*})=r^{*}{\left(1-r^{*}\right)}^{2n-1}.$$

Similarly, by differentiating *du**(*r*)/*dr** = 0, the maximum displacement of *u*_max_ is equal to 0.238 *u_o_* at *r** = 0.378. In this study, *n* = 2.0 is set to simplify the theoretical analysis. Substituting the derivatives of *w*(*r*) and *u*(*r*) into Equations (1) and (2), the bending strain energy (*U_b_*) and stretching energy (*U_S_*) can be integrated as (see Appendix A)
(9)$$U_{b}=\frac{8\pi Eh^{3}}{9(1-v^{2})}\left(\frac{w_{o}^{2}}{a^{2}}\right)$$
(10)$$U_{S}=\frac{\pi Eh}{\left(1-v^{2}\right)}\left(\frac{6u_{o}^{2}}{35}-\frac{4}{15}\frac{u_{o}w_{o}^{2}}{a}+\frac{32}{105}\frac{w_{o}^{4}}{a^{2}}+\frac{4v}{21}\frac{u_{o}w_{o}^{2}}{a}\right).$$

During deformation, the radial displacement satisfies the principles of bending energy minimization in achieving equilibrium. Hence, *d**Us*/*d**u_o_* = 0, and the relation of *u_o_* and *w_o_* is obtained by
(11)$$u_{o}=\frac{\left(7-5v\right)}{9}\frac{w_{o}^{2}}{a}.$$

Substituting Equation (11) into Equation (10), the stretching strain energy (*Us*) can be simplified as
(12)$$U_{S}=\frac{8\pi Eh}{63}\frac{\left(19-5v\right)}{\left(1-v\right)}\frac{w_{o}^{4}}{a^{2}}.$$

The external pneumatic energy due to pressure loading is integrated by
(13)$$U_{P}=\int_{0}^{2\pi }\int_{0}^{a}Pwrdrd\theta =\frac{\pi Pw_{o}a^{2}}{3}.$$

For the deformation mechanism of the membrane, total energy (*U_T_*) consists of the strain energy, stretching energy, and external pneumatic energy. Therefore,
(14)$$U_{T}=\frac{8\pi Eh^{3}}{9(1-v^{2})}\frac{w_{o}^{2}}{a^{2}}+\frac{2\pi Eh}{189}\frac{\left(19-5v\right)}{\left(1-v\right)}\frac{w_{o}^{4}}{a^{2}}-\frac{\pi Pw_{o}a^{2}}{3}.$$

Therefore, the derivative of the total energy approaches zero (i.e., *d**U_T_*/d*w_o_* = 0) as the membrane attains maximum deformation. Thus, the value of *w_o_*/*h* can be expressed as
(15)$$\frac{16}{3(1-v^{2})}\left(\frac{w_{o}}{h}\right)+\frac{8(19-5v)}{63(1-v)}{\left(\frac{w_{o}}{h}\right)}^{3}=\frac{Pa^{4}}{Eh^{4}}.$$

Substituting Poisson’s ratio of 0.5 into Equation (15) yields
(16)$$\frac{64}{9}\left(\frac{w_{o}}{h}\right)+\frac{264}{63}{\left(\frac{w_{o}}{h}\right)}^{3}=\frac{Pa^{4}}{Eh^{4}}.$$

Equation (15) is used to predict the maximum deformation of the membrane under parameters such as the membrane’s radius, thickness, Young’s modulus, and driving air pressure.

Given the large deflection of the membrane, i.e., $w_{o}/h\gg 1$, Equation (16) can be simplified as follows:(17)$$\frac{w_{o}}{a}=0.62{\left(\frac{Pa}{Eh}\right)}^{\frac{1}{3}}.$$

## 3. Materials and Methods

### 3.1. Design Principle

The major contribution of this study is to design and fabricate a new pneumatic valveless pump. The novelty of this pump lies in its design in which a thin membrane is layered and provided with two holes. A centric hole is 10 mm in diameter. In addition, a small hole is 1.5 mm in diameter to serve as an inlet port (or outlet port) for connecting to an inlet (or outlet) fluidic channel. An actuated membrane (or deflection membrane) is attached to the thin-film membrane, as shown in Figure 1a. This design leads to an actuated diaphragm and check valve designed with the same thin membrane. Therefore, the position of the inlet port (or outlet port) is crucial for the proposed valveless pump. According to the design principle, the inlet port (or outlet port) should be as close as possible to the rim of the actuated membrane (or deflected membrane). In the rim of the actuated membrane, the deformation of the diaphragm is close to zero. Hence, theoretical and numerical modeling needs to be conducted to determine the proper positions of the two ports. The pump body consists of a fluidic chamber and two fluidic channels. In other words, this membrane exhibits the functions of a reciprocating diaphragm and a check valve. Figure 1a shows an exploded view of the valveless pump. The pump module is made of seven-layer PDMS structures. The thickness of the air chamber and that of the fluidic chamber with diameters of 18.0 and 10.0 mm, respectively, are theoretically calculated to be 6.36 and 2.90 mm, respectively, by using Equation (17). Figure 1b shows a schematic side view of the valveless pump. The outline dimension of the valveless pump is 30.0 mm × 30.0 mm × 20.0 mm. The geometrical specifications of each layer are listed in Table 2 in detail.

Figure 2 displays schematically the structures of the valveless pump and demonstrates the working principle of the actuated membrane for two processes, namely, suction and compression. The fluidic chamber of the valveless pump is initially empty when the actuated membrane does not pneumatically activate. During the suction process, the actuated membrane is pneumatically deformed to the convex surface, and then, the liquid is introduced to pass through the inlet port and fill the fluidic chamber. At the same time, the deflected membrane is sucked upward to block the outlet port (Figure 2b). During the compression process, the actuated membrane is pneumatically deformed down to the concave surface to close the inlet port and prevent backflow. The compressed liquid within the fluidic chamber pushes the deflection membrane downward to open the outlet port. Thus, the liquid can be directly expelled through the outlet port (Figure 2c). Thus, the actuated membrane is able to perform suction and compression actions with a constantly driving frequency. The liquid is pumped from the inlet channel to the outlet channel to achieve transportation performance. Notably, the gas exhibits the compressible property; therefore, the proposed valveless pumps cannot be suitable for the gaseous working fluid.

### 3.2. Chip Fabrication

The valveless pump is laminated with seven-layer PDMS structures. The four main structures are the air chamber, diaphragm, thin membrane with an inlet/outlet port, and a fluidic chamber with two channels. The polymethylmethacrylate (PMMA) plates of the master molds are prepared by engraving them to form four main structures by using a CNC machine (EGX-400, Roland Inc., Hamamatsu, Japan). Then, a prepared PDMS is poured into the master molds to cast the inverse structures. The gaps between the actuated/deflected membrane and the thin membrane that contains the inlet port/outlet port both were 100 μm, which are not easy to control during the fabricating process. Finally, the seven-layer PDMS structures are sequentially bonded together by utilizing an oxygen plasma treatment to form the complete valveless pump. A close-up photograph of the fabricated valveless pump is shown in Figure 3.

### 3.3. Experimental Setup

The experimental platform was composed of an air compressor, a functional control circuit, and two electromagnetic valves (EMVs) (SMC Inc., s070 m-5bg-32, Taoyuan City, Taiwan) to evaluate the performance of the proposed pump as shown in Figure 4. The performance of EMVs is to close or open the input or output of compressed air. In addition, a digital controller is incorporated with two EMVs to operate the frequency of air input/output. A series of tests was conducted to measure the pumping rates associated with the driving frequency and the driving pneumatic pressure. During the experimental observations, the proposed pump was positioned in front of a digital camera (E-5P, Olympus, Tokyo, Japan) to acquire the flow motion of the liquid. The pumping effect was evaluated by measuring the weight of the liquid output within a period of 1.0 min by using an electronic balance (AB54-S, Mettler Toledo, Taipei City, Taiwan). Consequently, the measured weight of the liquid was converted to the volume flow rate assuming a constant water density (1.0 g/cm^3^).

## 4. Results and Discussion

### 4.1. Estimation of Membrane Deformation

To investigate the transportation performance of the valveless pump, it was fabricated using flexible structures that were pneumatically activated by an air chamber. A numerical simulation was performed to design the micropump and to investigate the deformation of the PDMS membrane. The deformation was simulated numerically using a commercial code (CFD-ACE+, CFD-RC, Huntsville, AL, USA). Enhanced first-order brick elements were recommended to be introduced to the mechanism of the PDMS membrane [20]. The moving boundary condition was simulated using the stress and deformation modules. The moving boundary of the membrane was discretely separated to ensure smooth motion. The deformation grid of the moving boundary was constructed using the auto-remesh function in the deformation process. The geometric dimension of the membrane was 18 mm in diameter and 0.3 mm in thickness. In this simulation, a 3D numerical domain was discretized into approximately 400,000 cells with structured hexahedral meshes. Dense grids were used in the moving boundary regions where deformation was induced by the air chamber. The convergence criteria of nonlinear stress residuals and shared residuals were set to be 10^−3^ and 10^−8^ respectively to ensure the numerical convergence during all of the simulations [20]. The density (ρ), Young’s modulus (E), and Poisson’s ratio (ν) of the PDMS membrane were 970 kg/m^3^, 1.4 MPa, and 0.499, respectively [20].

The deformation profile of the diaphragm was established to optimally design the position of the inlet or outlet port. To measure the deformation profiles (i.e., w), the pneumatically actuated membrane was placed sideways on a stage of a digital camera. The pneumatically actuated membrane with a thickness of 300 µm and diameter of 18.0 mm was applied with the adjusted static pressure. The experimental and numerical deformations of the flexible membrane under the different air pressures of 20.68, 34.47, 48.26, and 68.95 kPa are as shown in Figure 5 and Figure 6. In principle, the deformation of the actuated membrane increased with the increasing driving pressure. The increasing deformation of the flexible membrane yielded an increasing stroke volume to enhance the pumping performance. As observed in the results of the experiments and simulations, the deformation exhibited hemisphere-shaped profiles.

Figure 7 shows the comparisons of the theoretical, numerical, and measured data for the z- and r-directional maximum deformation of the membrane against different air pressures. The maximum deformation of membrane is located at *r* = 0 in the transversal direction and at *r* = 3.4 mm in the radial direction. The membrane of *w_max_* is designed to the thickness of the air or fluidic chamber. As expected, the z-directional maximum deformation increased with the increasing driving pressure. The increasing deformation of the flexible membrane yielded an increasing stroke volume to enhance the pumping performance. Notably, the relationship between the maximum deformation and the driving pressures was approximated by a one-third power law. The numerical simulations generated better results than the theoretical calculation for the maximum deformation of the membrane. However, the finite element method applied to the numerical model required extensive simulation time, and convergence was difficult to achieve in the calculation of the deformation behavior of the membrane. Hence, the theoretical analysis easily predicted the deformation profiles of the membrane. Furthermore, the theoretical and numerical models could be used to determine the proper locations of the inlet or outlet ports on a 300.0 μm-thick diaphragm. The theoretical calculations obviously exhibited reasonable agreement with the numerical simulations for the maximum z-directional displacements, especially under high driving pressure. However, the values obtained in the theoretical analysis and numerical simulation showed large variations for the maximum r-directional displacement. The deviation was 23.4% under a pressure of 68.95 kPa. Nevertheless, the theoretical predictions were consistent with the numerical calculations.

As mentioned previously, the position of the inlet port (or outlet port) is crucial for the valveless pump. The deformed size of the membrane affects the flow rate during inhalation or exhalation. Figure 8 shows the comparisons of the theoretical calculation and numerical simulation for the deformed profiles of *w**(*r**) and *u**(*r**). It was facilitated to determine the proper position of the inlet or outlet port. The profiles of the z- and r-directional displacements obtained by theoretical prediction agreed well with those obtained by the numerical calculation. These results revealed that the theoretical analysis successfully verified the feasibility of the large deformation of the membrane. Moreover, the deformation of the membrane was close to 0 as *r** > 0.9. Therefore, the inlet/outlet port was designed to be at the proper location of 0.5 < *r** < 0.9. In this study, the proper location of the inlet/outlet port was designed to be *r** = 0.8 (~1.8 mm) from the rim of the membrane with a diameter of 18.0 mm.

### 4.2. Characterization of the Pump

The pumping rate was investigated at different driving frequencies and operation modes under a driving pressure of 68.95 kPa. Experiments were performed to validate the pumping performance of the proposed pump. The suction and compression strokes of the diaphragm were verified and are demonstrated in Figure 9. The liquid could be inhaled from the inlet port to the fluidic chamber during the suction stroke (Figure 9a,b). On the contrary, the liquid could be expelled from the outlet port to the fluidic channel during the compression stroke (Figure 9c,d) (see Appendix A).

Finally, the valveless pump was verified to achieve high pumping performance. Figure 10 shows a series of photographs of the valveless pump at different times. The beakers shown in the left and right of the figure are first loaded with red ink and transparent water, and the proposed pump is placed over them to avoid the siphoned effect. The operating conditions were a driving frequency of 5.0 Hz and driving air pressure of 68.95 kPa. To ensure the accuracy of the measurement, the liquid was first pumped into the valveless pump to eliminate the internal bubbles. Figure 10a shows the initial condition where the liquid in the side tubes was transparent, and no significant transportation could be observed. When the diaphragm was activated by pneumatics, the fluidic tubes were filled with the red liquid. This result revealed that a pumping effect was induced by the suction and compression processes of the diaphragm. The images show that the valveless pump could achieve a pumping rate of 12.47 mL/min (Figure 10b–f)) (see Appendix A).

Sequentially, the operating modes were varied to examine the effect of the suction strokes (i.e., half-cycle stoke) and suction plus compression strokes (i.e., full-cycle stoke) for the different driving frequencies under the pressure of 69.85 kPa. During the process of suction plus compression in the lower frequency region (1–5 Hz), the flow rate increased, reached a peak value, and then decreased as the frequency increased. However, in the high frequency region (5–13 Hz), the flow rate decreased with increasing frequency. The same phenomena also occurred during the suction process, but the driving frequency with respect to the peak flow rate was different. The peak flow rate occurred at 7.0 Hz. The maximum flow rate is known to be limited by the time when the liquid within the fluidic chamber is inhaled and expelled. If the driving frequency is high, then the liquid cannot be completely suction or compression of the fluidic chamber. Therefore, the flow rate will not increase with increasing frequencies, but it will gradually decrease. These peak points are related to the characteristic properties of the suction or compression strokes, as clearly seen in Figure 11. The half and full cycle strokes achieved the maximum flow rates of 5.69 and 12.47 mL/min at frequencies of 7.0 and 5.0 Hz, respectively. The flow rate range of the pump is important for some applications; the flow stability is also critical while the fluctuation of the flow may cause an inaccuracy of the delivered volume. A stability test for the proposed pumps was performed at pressure of 68.95 kPa and frequency of 5.0 Hz. It showed the results that the flow rate slightly dropped by a variation of 8.9% after 60 min of operation, as shown in Figure 12.

## 5. Conclusions

In this study, a valveless pump was successfully designed and fabricated. A newly designed pump utilizes an actuated or a deflected membrane, and it serves as the function of a check valve at the same time. A theoretical model of the large deformation was derived to successfully predict the profiles of deformation and determine the proper location of the inlet or outlet port. Finally, the valveless pump was fabricated and experimentally investigated to determine its pumping performance. The flow rate could reach as high as 5.69 and 12.47 mL/min at frequencies of 5.0 and 7.0 Hz with respect to the half cycle and full cycle strokes, respectively, under the pressure of 68.95 kPa.

## Figures and Tables

**Figure 1 micromachines-13-00016-f001:**
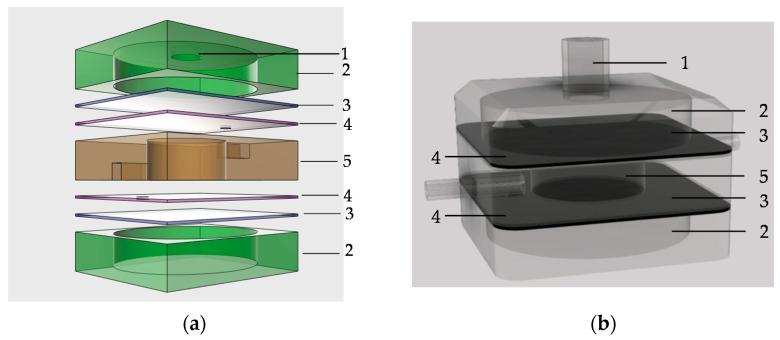
(**a**) Exploded view and (**b**) schematic entity of valveless pump.

**Figure 2 micromachines-13-00016-f002:**
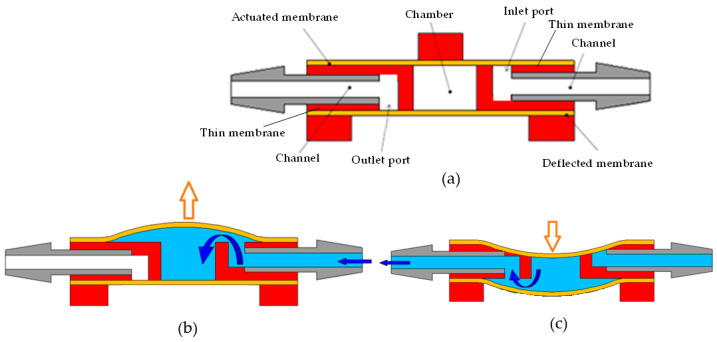
(**a**) Schematic front view of valveless pump and the working principle of the (**b**) suction process and (**c**) compression process.

**Figure 3 micromachines-13-00016-f003:**
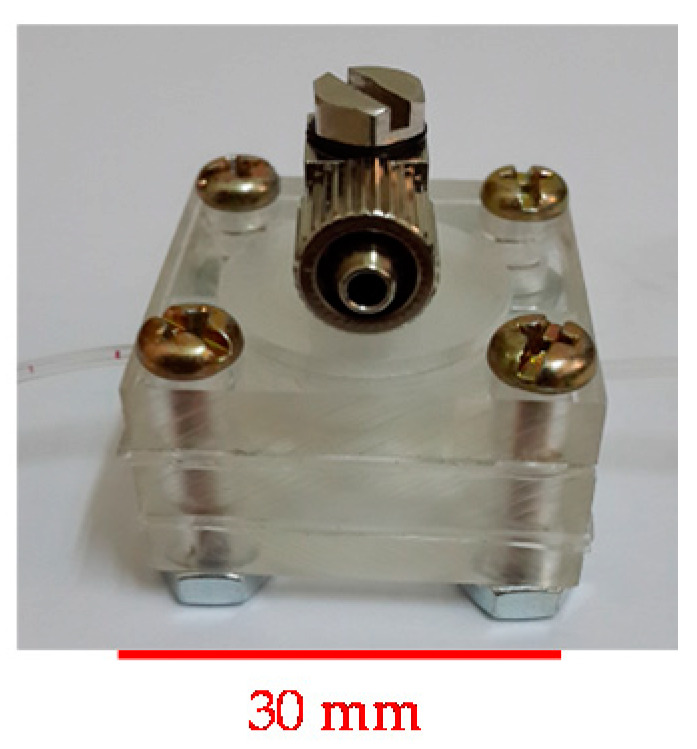
Close-up photograph of a valveless pump.

**Figure 4 micromachines-13-00016-f004:**
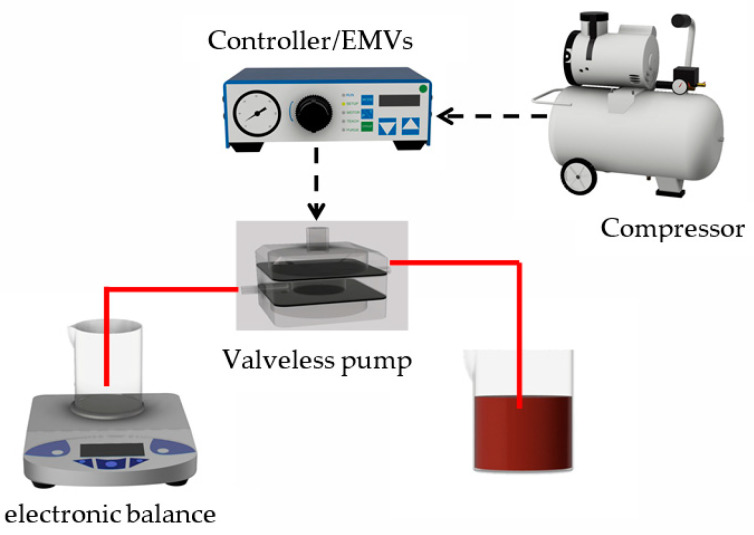
Schematic diagram of experimental setup.

**Figure 5 micromachines-13-00016-f005:**
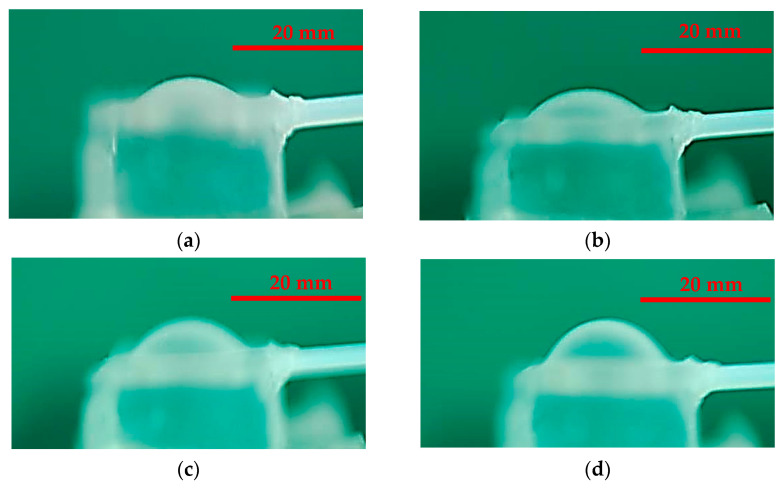
Experimental deformations of the actuated membrane under the pressures of (**a**) 20.68 kPa, (**b**) 34.47 kPa, (**c**) 48.26 kPa, and (**d**) 68.95 kPa.

**Figure 6 micromachines-13-00016-f006:**
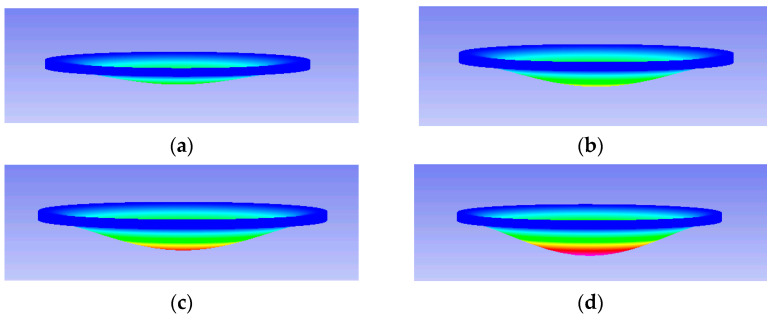
Numerical deformations of the actuated membrane under the pressures of (**a**) 20.68 kPa, (**b**) 34.47 kPa, (**c**) 48.26 kPa, (**d**) 68.95 kPa, and (**e**) deformation contour legend.

**Figure 7 micromachines-13-00016-f007:**
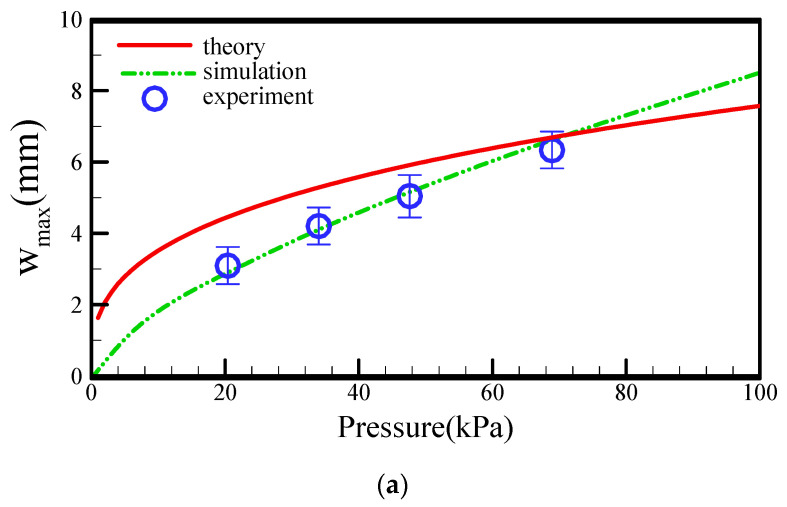
Comparisons of theoretical, numerical, and measured data for (**a**) w_max_ and (**b**) u_max_ with a diameter of 18.0 mm against different air pressures.

**Figure 8 micromachines-13-00016-f008:**
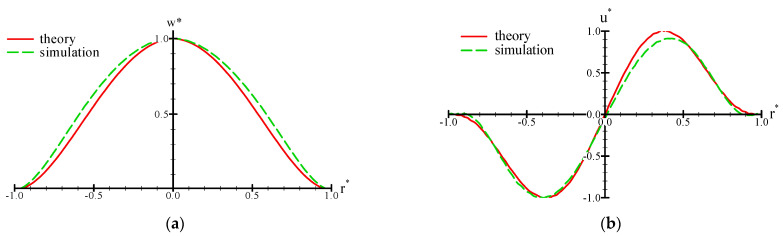
Comparisons of theoretical calculation and numerical simulation for the profiles of (**a**) *w**(*r**) and (**b**) *u**(*r**). Here, *w** = *w*/*w*_amx_, *u** = *u*/*u*_max_, and *r** = *r*/*a*.

**Figure 9 micromachines-13-00016-f009:**
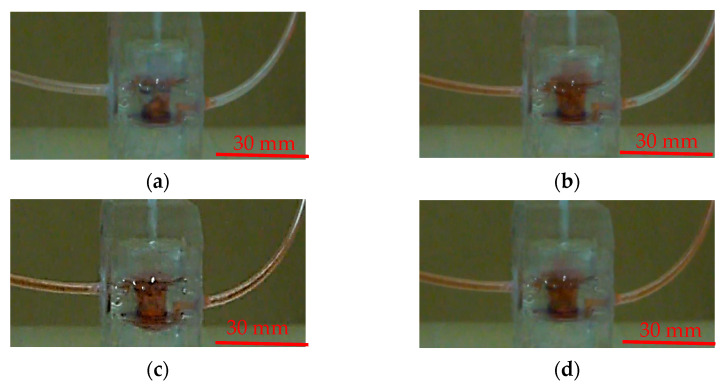
Photograph demonstrating the (**a**) initial-state, (**b**) suction, (**c**) compression and (**d**) expel of the valveless pump.

**Figure 10 micromachines-13-00016-f010:**
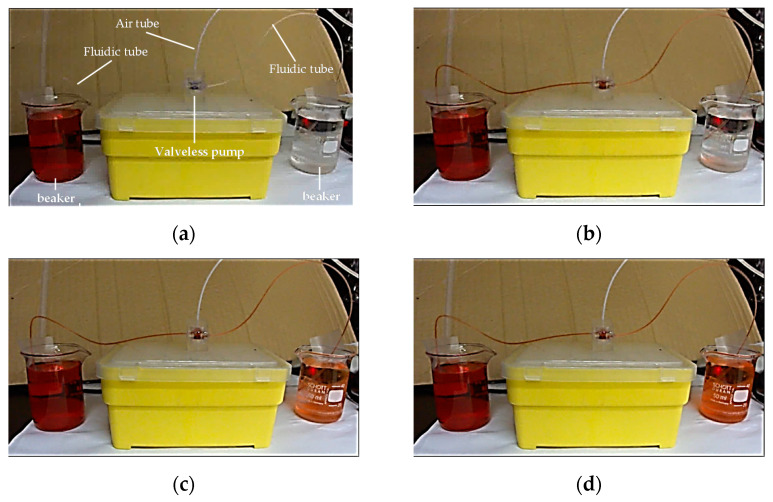
Series of photographs of valveless pump showing pumping performance at different times. (**a**) t = 0 s; (**b**) t = 10 s; (**c**) t = 20 s; (**d**) t = 30 s; (**e**) t = 40 s; and (**f**) t = 60 s.

**Figure 11 micromachines-13-00016-f011:**
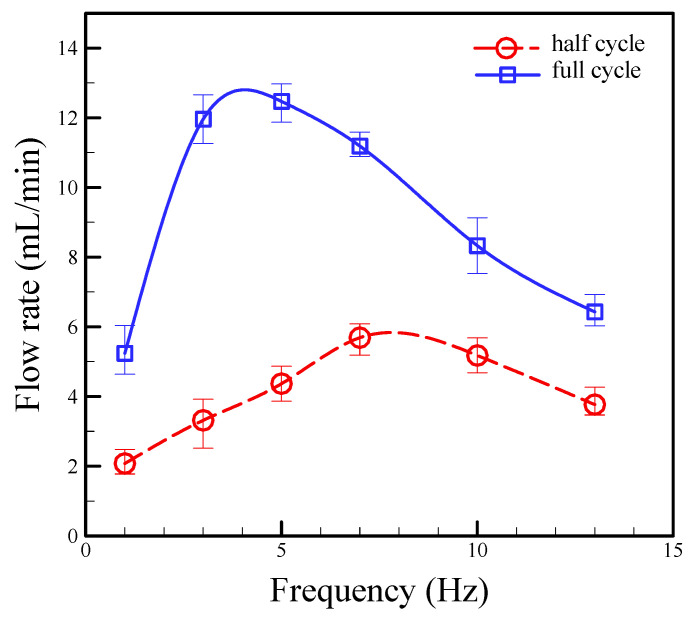
Relationship between flow rate and operating frequency at different operation modes of half cycle and full cycle strokes. Error bars were evaluated by three samples and six tests.

**Figure 12 micromachines-13-00016-f012:**
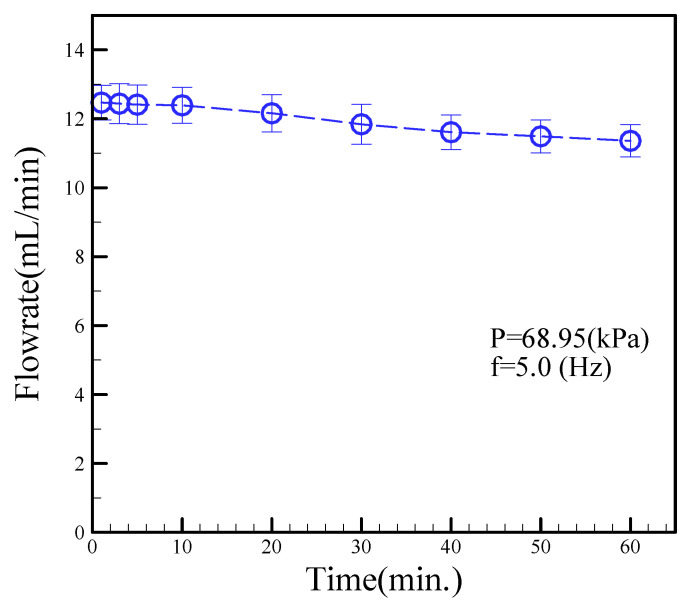
Valveless pumps test stability at the pressure of 68.95 kPa and frequency of 5.0 Hz. Error bars were evaluated by one sample and three tests.

**Table 1 micromachines-13-00016-t001:** Specifications and performance comparisons of valveless pumps.

First Author	Year	Actuation	Type	V_max_ (V)	P_max_ (kPa)	f_Qmax_ (Hz)	Q_max_ (mL/min)	Ref.
Hasan	2017	MHD-DC					22.64	[9]
Russel	2016	EHD-DC		700	6.7		0.47	
Tawfik	2017	EO-DC		10			0.08	
Hu	2017	Electrowetting		9		100	0.1	
Wang	2017	Electrochemical	Diffuser	2		0.5	0.766	
Liu	2016	Bubble			40		0.07688	
Kim	2015	Electrostatic	Peristaltic		17.5	17,000	4.0	[21]
Shen	2011	Motor/magnetic	Peristaltic		6.6	12	2.4	
Nakahara	2013	Piezoelectric	Peristaltic	140		210	1.5	
Cui	2011	Pneumatic	Peristaltic		0.79	30	0.6	
Sun	2008	Thermal-SMA	Peristaltic	2.5		0.3	0.9	
Izzo	2007	Piezoelectric	Tesla	100	1.73	2250/3000	0.64	[25]
Tseng	2019	Piezoelectric	Diffuser/nozzle	160	2.9	400	0.4	

**Table 2 micromachines-13-00016-t002:** Geometrical specifications of valveless pump.

Index	Section	Geometrical Dimension			
		Diameter (mm)	Length (mm)	Width (mm)	Thickness (mm)
1	air inlet port	1.5	-	-	1.0
2	air chamber	18.0	-	-	6.5
3	diaphragm	-	20.0	20.0	0.3
4	thin membrane	10.0/1.5	20.0	20.0	0.3
5	fluidic chamber	10.0	-	-	3.0
	fluidic channel	-	5.0	1.5	1.5

## Data Availability

The data presented in this study are available within the article. Other data that support the findings of this study are available upon request from the corresponding author.

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
