# Peer review of "Design and Fabrication of Pneumatically Actuated Valveless Pumps"

_micromachines, 2021, doi:10.3390/mi13010016_

Round 1

Reviewer 1 Report

Figure 1 is in page 4, but its title is in page 5. Please arrange to have it in the same page.

Figure 1(b) should have some labels.

Figure 2 is in page 5, but its title is in page 6.

Figure 3 should replace with clearer picture.

3.3 Experimental setup: Should have schematic picture to help reader to understand more on how the experiment conducted.

Figure 4 is in page 7, but its title is in page 8. Please arrange it in the same page.

Figure 4 (a) are so blurry. Please replace with clear picture. 

What is the performance result of evaluating using EMVs?

Reviewer 2 Report

In this paper entitled “Design and Fabrication of Pneumatically Actuated Valveless Pumps” a new method for valveless pumping was proposed, along with numerical simulation and analytical approach. The paper introduces the ideation, numerical/analytical methods to design, fabrication, and experimental results for this valveless pump. The flow is relatively clear and the text is well-written except for a couple of grammatical issues. I recommend major revision, and below are my comments:

Major comments:

  1. The motivation of the paper is unclear. The introduction should be edited to motivate the publication, with specific examples of the application of the device and how a “valveless” micropump can be useful.
  2. Peristaltic micropumps are not mentioned in the introduction which are a significant category of mechanical micorpumps. These pumps rectify by actuators not valves (valveless), which is why they should be mentioned in the introduction and the current work should be distinguished from them.
  3. The paper should mention other papers in the literature on valveless micropumps (e.g. Caterpillar locomotion-inspired valveless pneumatic micropump using a single teardrop-shaped elastomeric membrane) and distinguish this device from them and clarify the contribution.
  4. Please provide reference or evidence for this sentence in paragraph 4: “The benefits of dynamic valves in micropumps include high operating pressure, good flow direction, and minimal backflow”.
  5. In the last paragraph of the introduction, it is mentioned that “… that utilizes an actuated membrane with the check valves in the same membrane.”, but the valve mechanism is not in the membranes. Please rewrite and clarify. Also, the outlet valve works based on the interaction between the outlet port and the deflected membrane, which is not the same as the actuated Please rewrite and clarify.
  6. In section 2 the only reference is [26] which is Timoshenko’s Theory of plates and shells. The papers should provide a brief literature review of the previous theoretical approaches for pneumatic micropumps and clarify what distinguishes the theoretical method presented in the paper. This could be a short paragraph in section 2.
  7. In the first paragraph of section 4.1 it is mentioned that in the simulation the Poisson ratio of PDMS is considered 0.5. For isotropic linear elastic solid materials, a Poisson ratio of 0.5 results in singular stress, and the simulation should not converge. Please clarify how the simulation was performed and converged.
  8. The micropump should be compared with similar micropumps in the literature in the same category (e.g., valveless, or pneumatic) to see the contribution. Further, it should be compared with other pumps on the footprint since it’s critical for the versatility of the device. Please add a table in section 4 for this comparison.
  9. The “deflected membrane” deflects due to the incompressibility of the working fluid, which suggests that the device cannot be a good fit for the gaseous working fluid. This should be briefly discussed in the paper as a limitation of the device.
  10. Figure 7 should be improved, the point is not clear. If it is all about the deflection of the membrane, it is not significant enough to have a specific picture for it. If it intends to show pumping, the fluid in the downstream should be partially filled and the picture should show displacement of the fluid in a cycle.
  11. Figure 8 is unclear, please provide more clear pictures and add labels in the picture itself.
  12. Figure 5 and 9 have error bars for the experimental section. How many devices have been made and how many of them have been tested? This should be clearly mentioned in the caption of the figures.
  13. If the results of the experimental study are for one sample, it does not provide enough data for the experimental results. Further devices should be built and tested (at least N=4).

Minor comments:

  1. In the last sentence of the abstract “applied value” should change to “driving pneumatic pressure” to avoid any confusion. This could be also used elsewhere in the paper.
  2. Please provide higher-quality pictures for Figure 4b.
  3. In section 4 it is mentioned: “Enhanced first-order brick elements were recommended to be introduced to the mechanism of the PDMS membrane [33].” There is no [33], please fix it.

Reviewer 3 Report

In this paper, the author proposed a new design and fabrication process for pneumatically actuated valveless pumps. The performance of this pump looks great and would have large potential to be used in several fields.

Overall, I find the interest in this research work. However, there is major points that I think the authors need to pay additional attention before the publication of this work.
Below are my specific concerns.

1. it would be better if the author could discuss more about its potential applications in this paper.

2. Some other types of pumps should also be include in the introduction section, such as electroosmosis pump, osmosis pump, etc. The introduction should be refined and the novelty should be emphasized. I suggest the author could add a table to compare the performance of the various pumps.

3. The author defined the proposed pump as a valveless pump. But in the context, check valve has been used to describe this pump, it may cause confused to readers. It should be clarified more carefully.

4. Although the flow rate range of the pump is important for some applications, the flow stability should also be critical and fluctuate of the flow may cause inaccuracy of the delivered volume. I suggest the author could add some data and discuss with regard to the flow stability along the time.

5. Could the author specify the differences between theory and simulation in Fig. 5 and 6 in more detail?

Reviewer 4 Report

This paper is interesting and worthwhile. In this paper, the authors describe a valveless pump for medium transportation. I think this paper can be published if the authors could address the following minor problems properly.

  1. In this paper, the authors are mainly focusing on mechanical pumps with moving parts. I think the authors should mention other non-mechanical pumps in the introduction, for example, EHD pumps or hydrogel pumps.

Yoshimura, Kyosuke, et al. "Autonomous oil flow generated by self-oscillating polymer gels." Scientific Reports 10.1 (2020): 1-7.

Mao, Zebing, Takeshi Iizuka, and Shingo Maeda. "Bidirectional electrohydrodynamic pump with high symmetrical performance and its application to a tube actuator." Sensors and Actuators A: Physical 332 (2021): 113168.

  1. It is better to show the gaps between the actuated membrane and inlet port/outlet port, which are not easy to control during the fabricating process.

  1. Please add a scale bar in Fig.3 and Fig.7. Use a clear photo for Fig.3.

Round 2

Reviewer 3 Report

The manuscript has been improved, and it is suitable for publishing.